# Percentiles and Reference Values for Accelerometric Gait Assessment in Women Aged 50–80 Years

**DOI:** 10.3390/brainsci10110832

**Published:** 2020-11-09

**Authors:** Raquel Leirós-Rodríguez, Jesús García-Liñeira, Anxela Soto-Rodríguez, Jose L. García-Soidán

**Affiliations:** 1Faculty of Health Sciences, Nursing and Physiotherapy Department, Universidad de León, 24401 León, Spain; 2Faculty of Education and Sport Sciences, Campus a Xunqueira, University of Vigo, s/n, 36005 Pontevedra, Spainjlsoidan@uvigo.es (J.L.G.-S.); 3Health Service from Galicia (SERGAS), Galician Health Services—Ourense Hospital, s/n, 32005 Ourense, Spain; anxelasoro@hotmail.es

**Keywords:** gait, accidental falls, geriatric assessment, public health, womens health

## Abstract

*Background*: The identification of factors that alter postural stability is fundamental in the design of interventions to maintain independence and mobility. This is especially important for women because of their longer life expectancy and higher incidence of falls than in men. We constructed the percentile box charts and determined the values of reference for the accelerometric assessment of the gait in women. *Methods*: We used a cross-sectional study with 1096 healthy adult women, who were asked to walk a distance of 20 m three times. *Results*: In all of the variables, a reduction in the magnitude of accelerations was detected as the age of the group advanced. The box charts show the amplitude of the interquartile ranges, which increases as the age of the participants advances. In addition, the interquartile ranges were greater in the variables that refer to the maximum values of the accelerations. *Conclusions*: The values obtained can be used to assess changes in gait due to aging, trauma and orthopaedic alterations that may alter postural stability and neurodegenerative processes that increase the risk of falling.

## 1. Introduction

Each year, one in three adults over 65 years of age and one in two of those over 80 years of age will suffer a fall [1]. Thirty percent of falls cause moderate or severe injuries, and in elderly people, falls result in fractures, functional deterioration, reduction of physical activity levels, premature entry into residential care institutions, fear of falling and even death [2,3]. Identifying factors that alter postural stability is fundamental to the design of interventions to maintain independence and mobility, and is especially important in the case of women due to their longer life expectancy and higher incidence of falls [4].

Research studies have mostly based their results on analyses conducted with force platforms and an electronic walkway. This tool provides results based on the behaviour of the centre of pressure (CP) of the body. This parameter has been linked to the risk of falling, but it is not a reflection of the overall performance of the body in space [5]. On the other hand, there are also three-dimensional kinematic gait measurements. However, Mc Ginley et al. [6] concluded that, although most errors in gait analysis are probably acceptable, they are generally not small enough to be ignored during clinical data interpretation. A goal of any clinical measurement technology must be to provide measurements that are free from any measurement error that might affect interpretation. However, the most valid and sensitive instruments depend heavily on the skill of assessors in accurately placing markers [7].

An alternative, low-cost, portable method that is easy to apply to the analysis of cinematic movements of the individual is the use of accelerometers. These can quantify the movements of any body segment and, for the study of equilibrium, are fundamentally based on the behaviour of the individual’s centre of mass. Previous studies showed the sensitivity of these devices to small changes in dynamic postural control systems [8]. Gait analysis based on the study of the acceleration of the body has been a valid and reliable method of predicting the risk of falling or discerning population subgroups [9]. The study of body kinematics facilitates the detection of alterations in the gait early, when they are not yet detectable through visual analysis [10].

Dynamic postural control is related to the centre of mass (CM), which, according to Mapelli et al. [11], is the result of the multi-segmental conception of balance. The body can be conceived as a system of rigid bodies whose centre of gravity is the average of the centres of mass of all its segments, a definition that is along the lines proposed by Hogdes et al. [12]. It follows that CM control is one of the prerequisites for the maintenance of balance during activities of daily life, which include walking, going up and down stairs, stooping, sitting and standing [13]. Gait has traditionally been evaluated qualitatively (and, in many cases, subjectively) in clinical settings and quantitatively, using sensitive electronic gateways or photoelectric cells in laboratory environments. The systems used in laboratory environments provide data on temporal–temporal variables such as speed, cadence, length, variability and duration of stride and support time [14]. They compare the displacement of the CP between the feet, an independent parameter to the CM. That is, it is a parameter strongly conditioned by the intrinsic activity of the ankle and object of study with the inverted equilibrium pendulum theory [15]. However, this theory does not provide a complete account of the functioning of the dynamic postural control system and of all the strategies that it uses to maintain balance [16]. 

A recent systematic review identified that the use of accelerometers has the potential to positively influence interventions based on physical exercise to improve balance and prevent falls in older people [17,18]. However, a standardized evaluation protocol or reference accelerometric values have not been defined. Therefore, this investigation was carried out with the aim of constructing the percentile box charts and determining reference values for accelerometric assessment of gait in adult and older women, with the initial hypothesis that the reference values are different as age advances and, consequently, with the aging process.

## 2. Materials and Methods

### 2.1. Sample

A descriptive and cross-sectional study was carried out in a random sample of adult women from the city of Ourense (Spain) during the months of March and April 2019. All of them were recruited from municipal sports centers. The city of Ourense had a population of 57,543 women in 2018. To reach a confidence level of 95% and a margin of error of 3%, the participation of 148 women was established, and 1232 women were contacted and agreed to participate in the study. After the initial evaluation, 1096 (1.9% from the total female population) were selected, screened and found eligible, and 88.6% enrolled.

The following inclusion criteria were used: (a) engaged in physical activity between one and two days/week; and (b) walked between 30 and 90 min four days a week. The exclusion criteria were: (a) the inability to walk independently; (b) use of external orthopaedic elements to maintain bipedal static balance with eyes open for 60 s; (c) the presence of any contraindication or illness that prevented undergoing any of the tests; and (d) the antecedent of having fallen in the last year. This procedure is detailed in Figure 1.

### 2.2. Procedure

A triaxial accelerometer (GT3X+ ActiGraph LLC, Pensacola, Florida, USA) was used for the measurement of acceleration. This accelerometer allows a time series of acceleration data to be stored in a non-volatile flash memory. The small dimensions of these devices (4.6 cm × 3.3 cm × 1.5 cm), their low weight (19 g), accuracy (3mg/LSB) and a range of ± 6 units of gravity (g) make them a good choice to evaluate body position changes in outpatient environments.

This accelerometer provides accelerometric data in all three axes: axis 1 corresponds to the acceleration in the vertical axis (VT); axis 2, to the mediolateral (ML); axis 3, to the anteroposterior (AP), and the root mean square (RMS) of them. All accelerometers used in the study were calibrated static before use. The accelerometer measurements were configured for a time frame of 1 s. The sampling frequency selected was 50 Hz. Then, the signal was processed with a 30 Hz filter before being analysed [17,18]. This threshold is effective to eliminate the noise of the signal. The noise can come from the recording system itself if it is not properly attached (an aspect that must be solved with the previous calibration of the device and its proper fixing). Another origin of noise may be the selected sampling frequency, which should not exceed 50 Hz for the study of human movement nor be too low, which may skew data collection [19,20].

During testing, the women wore socks and comfortable clothing so that they could perform the tests comfortably. The accelerometer was placed directly on the skin at the height of the spinous process of the fourth lumbar vertebra. The device was secured with an adjustable belt and hypoallergenic adhesive tape to ensure that it did not move independently to the woman’s trunk during the test. The participants were asked to walk a distance of 20 m, divided into two sections of out and back, three times. The beginning and end points of rotation of the turning were properly marked (with a 50 cm high plastic cone at each end of the run). The tests were separated by intervals of 30 s to prevent the effects of lower limb muscle fatigue [21]. The analysis of results was based on the study of the average of the accelerations in the three attempts. 

In accordance with the Declaration of Helsinki (rev. 2013), all participants signed informed consent prior to their participation in the study. The institutional review board approved the study protocol and granted the ethical approval from the Commission of Ethics of the Faculty of Sciences of Education and Sport of the University of Vigo (Spain) (code: 3-0406-14).

### 2.3. Statistical Analysis

For the analysis of the results, the sample was divided into six age groups: G1, between 51 and 55 years (*n* = 187); G2, between 56 and 60 years (*n* = 172); G3, between 61 and 65 years (*n* = 185); G4, between 66 and 70 (*n* = 192); G5, between 71 and 75 years (*n* = 187); and G6, between 76 and 80 years (*n* = 173). The analysis of variance (ANOVA) with the Bonferroni correction was used to determine whether the differences between the groups were significant.

For the construction of the box charts and the calculation of accelerometric reference values, the chronological age of the participants was established as the explanatory variable (years), and the accelerations recorded as the response variable (gravitational unit or g). 

In order to obtain more accurate accelerometric data, a wide range of percentiles was established for the response variable, taking the proposal included in the study for the development of growth standards in children of the WHO Multicentre Grow Reference Study Group as a model [22]. Extreme outliers were removed from the sample according to the criterion x < Q (25) − 3 * IQR and x < Q (75) + 3 * IQR (where IQR is the interquartile range) so as not to excessively affect the most extreme percentiles of the distributions.

For the evaluation of normality and homoscedasticity, the hypothesis tests of Kolmogorov–Smirnov and Levene were used, respectively. The Mann–Whitney U test was used to confirm the observed differences in results between age groups. This analysis was performed using the IBM SPSS Statistics for Macintosh software, Version 20.0 (SPSS, an IBM Company, Armon, NY, USA).

For the construction of the percentile box charts and the calculation of reference values in each group, Generalised Additive Models of Position, Scale and Form (GAMLSS) were applied [23]. The data distributions of the response variable (acceleration) were modelled by exponential Box–Cox power distributions (BCPD), applying a cubic splines technique as a smoothing method and the worm plots [22] for the evaluation of the goodness of the adjustment. To carry out this analysis, the “gamlss” package of the statistical software R (R Core Team, 2014) was used.

In this study, to assess and control for potential confounding, it was taken into account that all selected women had the same active lifestyle (similar levels of physical activity practice), comorbidities or medication use and no orthopedic, traumatological or neurodegenerative pathology that presented with impaired balance and stability of gait. Criteria used to identify and include confounding variables in multivariable models were theoretical and based on consistent prior findings in the literature of confounding associations between each variable with the outcomes. Interactions were conducted post hoc through logistic regression analysis to examine the effects of (1) weight, (2) body mass index, and (3) age. These tests were conducted to evaluate the roles of age and obesity in explaining the group differences observed. All continuous variables were centered around their means prior to computing interaction terms.

## 3. Results

Participants’ descriptive data are summarised in Table 1, which shows that weight increased with age, whereas height decreased, and thus body mass index increased with age.

Data on equilibrium (accelerations) in the three spatial axes are shown in Table 2. All of the variables showed a reduction in the magnitude of accelerations with age. In the six study groups, the null hypotheses of normal distribution (*p* < 0.01) and homoscedasticity (*p* < 0.01) of accelerations were rejected. Likewise, the kurtosis values of the distributions determined values of positive asymmetry (>0.5) and leptokurtosis (>0.5) in all groups and variables groups. The average duration of the three attempts test presented by age groups is shown in Table 3, which can identify how the time it took to pass the test proceeded with advancing age.

There were age differences in balance (*p* < 0.01). In addition, the IQR analysis revealed that this increased with age; in other words, the variability in dynamic balance (during gait) also increased with age.

The graphs of acceleration percentiles for women on the walking test throughout aging are presented in Figure 2. This figure shows the amplitude of the IQR, which increased with the age of the women participants. In addition, the IQR was greater in the variables that refer to the vertical axis and the root mean square of the accelerations (in comparison to the mid-lateral and anterior-posterior axes). In the same way, the average duration of the three attempts of the gait test was significantly increasing as the age of the participants increased (Figure 3).

The cash graphs show similar trends in all groups. The magnitude and amplitude of the boxes increased with age.

Variables included in the logistic regression models were age, weight and BMI. Outcomes were the mean and maximum values of vertical, mid-lateral and anterior-posterior axes and the mean and maximum values of RMS. In our logistic regression models, the percentage of concordance ranged from 72.3–76.5%, supporting overall model fit. In multiple logistic regressions, older or heavier women were more likely to have minor accelerations (56.9%) than younger (41.7%), with an OR = 1.6 (95% CI = 1.1–2.4), or lighter women (44.2%), with an OR = 1.8 (95% CI = 1.2–2.5). The effect of presenting higher BMI was the same as having more weight, but with a smaller effect on the result of the model.

## 4. Discussion

In this study, we calculated percentiles and reference values for accelerometric assessment of gait in elderly women, i.e. women aged between 51 and 80 years. We split the sample into groups (G1 to G6) for later comparisons. The results provide the first normative values for this evaluation procedure, which is already widely used in a research environment, but not in clinical practice. The large sample size and use of a well-established evaluation procedure mean that our results can be considered representative. It should be noted that the reference values obtained are subject to the eligibility conditions of the participants of this work: healthy women with an active lifestyle and without trauma or orthopaedic conditioners. Set conditions indicate that the results obtained show the expected results in the study of balance during gait of a healthy adult or older woman. We have highlighted differences between the accelerometric variables defining the gait of specific groups, but these group comparisons should be treated with caution due to the modest size of the groups.

To date, no reference values have been published for accelerometric evaluation of walking. This gap in the data has hindered the development and dissemination of this technique, encouraging health professionals to continue using evaluation scales (Timed Up & Go, Berg Balance Score, ABC scale, amongst others) that include a multitude of tests, yet are not particularly sensitive to premature deterioration of balance. The existing literature on accelerometric gait analysis emphasises the importance of the values of the acceleration module (RMS) and of accelerations in the sagittal plane. Both have been strongly associated with risk of early falls [24], such that higher values of these acceleration parameters are related to greater lower limb strength and lower total percentage of fat by mass and lower limbs in older people (especially from 70 years old).

The study of the acceleration module is a constant in accelerometry research. This measure of the magnitude of movement has been used in virtually all accelerometric studies since this method of assessing balance and progress was first introduced [10,25,26].

Regarding movements in the sagittal plane, previous studies have reported exaggerated balancing in the mid-lateral axis during walking, along with compensations associated with deterioration [27]. This deterioration, in particular, would be due to rigidity of the pelvic girdle, and would break with the physiological premise of the energy economy principle. In addition, aging reduces the mobility of the lower limbs in the sagittal plane, and this is compensated for by an increase in flexion extension movements (horizontal plane) [28,29]. These compensatory movements in the horizontal plane are encouraged by excess fat mass [30].

In all groups, walking speed (calculated from running test time) and the magnitude of the accelerations recorded decreased with age. Accelerations and gait speed declined with age as a result of an increase in cadence and a reduction in step length [31]. These data are consistent with previous observations that walking speed declines with age, even in the absence of pathology [32]. In addition, even G6 retained the capacity to respond efficiently to disturbances, as accelerations did not increase in any of the axes [33]. 

The results indicate that during normal aging, walking speed and accelerations during walk decline. The reduction in the speed of walk makes falls more likely, but it serves to preserve stability in the face of age-related alterations in neuromotor, muscle and proprioceptive functions [34]. However, if the recorded accelerations do not increase, the sequence of the gait has not been altered [24]. The relevance of the data presented in this study lies in the fact that they make it possible to define cut-off points or reference values for gait stability in older women. Such cut-off points are essential to the design of clinical or epidemiological studies and useful even in clinical practice

It should be noted that the reference values we obtained apply only to women meeting the same criteria as our participants: good health, an active lifestyle and an absence of trauma and orthopaedic conditioners, a set of conditions that indicate that the results obtained show the expected results in the study of the balance of a healthy adult or older woman. 

The reference values obtained can then be used to determine whether the gait of a woman is within the normal range. Likewise, the percentiles allow one to make medium- and long-term predictions about functionality and balance, which can be useful for predicting the risk of falling at certain ages. Furthermore, the percentiles provided can (and should) be used by health professionals responsible for the treatment and management of patients with neurological diseases (doctors, physiotherapists, occupational therapists and nurses, mainly) to evaluate the degree of deterioration of balance of their patients, identify the degree of instability during walking, plan the most appropriate treatment (thus improving decision-making) and evaluate the results of therapeutic interventions carried out. Simultaneously, the accelerometric reference values presented here can be used for the early detection of neurological pathologies that present with altered postural control (such as dementia or Alzheimer’s, Huntington’s and Parkinson’s diseases) [35,36].

The main limitations of this study are its cross-sectional design, the absence of men from the sample and the lack of data on middle-aged adults. The most important interaction variable is the specific physical activity carried out by each of the participants and the existing variability in their lifestyles, with heterogeneous security as a consequence of the open nature of the inclusion criteria. It should be noted that the strict criteria of inclusion and exclusion selection of the sample employs at the same time, which ensures the reliability and validity of the results, and limits their generalizability to the total adult population and older women. Finally, the results of this work cannot be applied in assessments that do not follow the measurement protocol used in this research.

## 5. Conclusions

This study is the first time that percentiles and reference data for the accelerometric assessment of gait in adult and older women have been presented. These data enable independence and stability in walking and risk of falling to be assessed more precisely than is possible using clinical tests of balance. The values obtained can be used to assess changes in gait due to aging, trauma and orthopaedic alterations that may alter postural stability and neurodegenerative processes that increase the risk of falling. In addition, the possibility of projecting the results could be used to improve the quality of medical treatments and physiotherapy aimed at improving balance.

The relevance of the data presented in this study lies in the fact that it is possible to establish cut-off points or reference values for balance in older women from now on. These cut-off points are essential for designing clinical or epidemiological studies and even for their application in daily clinical practice.

## Figures and Tables

**Figure 1 brainsci-10-00832-f001:**
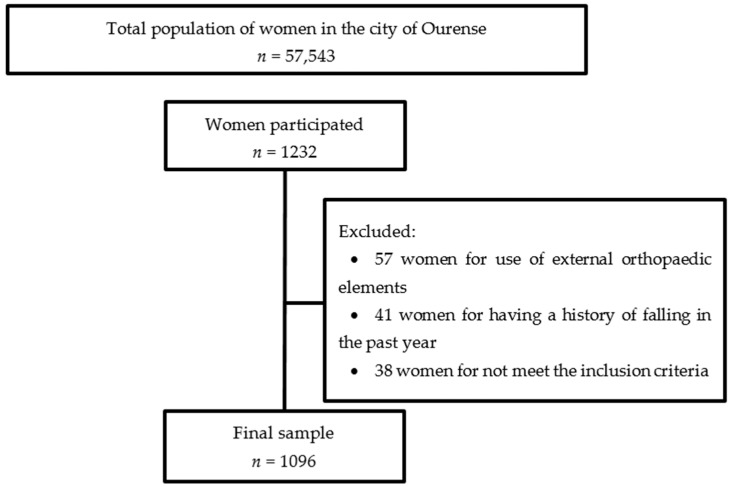
CONSORT flow diagram.

**Figure 2 brainsci-10-00832-f002:**
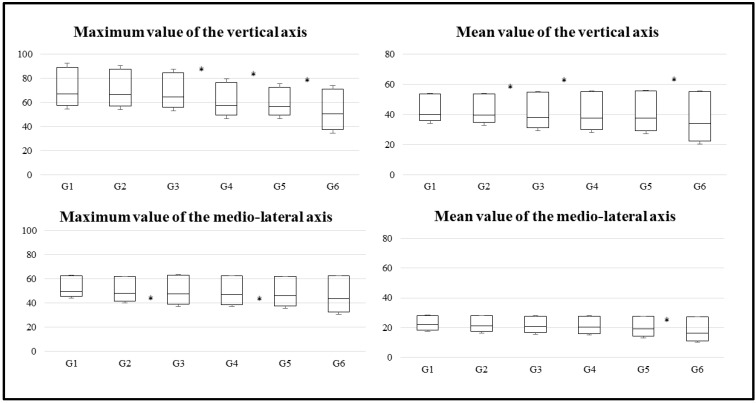
Box charts for the three axes and root mean square by age group. * Indicates statistically significant differences (*p* < 0.05) between the groups.

**Figure 3 brainsci-10-00832-f003:**
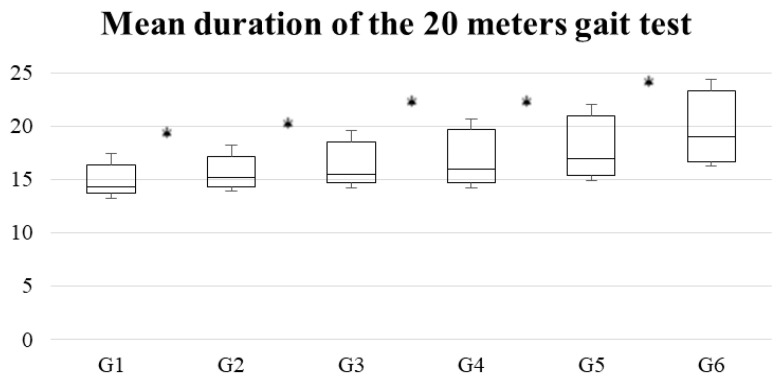
Box chart for the duration of the gait test by age group. * Indicates statistically significant differences (*p* < 0.05) between the groups in which it is found.

**Table 1 brainsci-10-00832-t001:** Descriptive statistics of anthropometric variables (data provided: mean ± standard deviation).

Age Group	N	Age (Years)	Weight (kg)	Height (cm)	Body Mass Index (kg/m^2^)
All	1096	68.8 ± 10.4	65.6 ±10.1	153.9 ± 5.4	27.6 ± 4.1
G1 (51–55 years)	187	53.4 ± 4.4	63 ± 7.6	155.6 ± 5	26 ± 3.3
G2 (56–60 years)	172	57.4 ± 4.3	64 ± 6.5	154.6 ± 6	26.8 ± 5.3
G3 (61–65 years)	185	64.2 ± 2.7	66.4 ± 11.1	154 ± 5.5	28 ± 4.7
G4 (66–70 years)	192	68.4 ± 3.8	63.8 ± 9.6	152.9 ± 6.1	29 ± 6.3
G5 (71–75 years)	187	74.2 ± 4.6	66.5 ± 10.1	151.8 ± 5.2	28.3 ± 3.2
G6 (76–80 years)	173	77.6 ± 2.2	68.1 ± 11.7	151.3 ± 4.2	29.2 ± 1.8

**Table 2 brainsci-10-00832-t002:** Percentiles and descriptive statistics (in g) for the three axes and root mean square by age group.

Variable	G1 (*n* = 187)	G2 (*n* = 172)	G3 (*n* = 185)	G4 (*n* = 192)	G5 (*n* = 187)	G6 (*n* = 173)
**Maximum value of vertical axis**						
Mean ± standard deviation	67.7 ± 17.4	67.1 ± 19.8	63.7 ± 16.3	56.8 ± 13	57.3 ± 11.4	51.3 ± 14.1
Kurtosis	3.9	3	2.9	3.6	3.4	1.7
Percentile 25	57.3	57	56	49.7	49.3	37.7
Percentile 50 (median)	67	66.3	64.3	57.3	56.3	50.7
Percentile 75	79.3	78	76	68.7	65.3	57.7
Interquartile range	22	21	20	19	16	20
**Mean value of vertical axis**						
Mean ± standard deviation	44.1 ± 13.7	41.9 ± 13.7	41.1 ± 12.6	35.9 ± 10	36 ± 9.6	32.7 ± 11.1
Kurtosis	3.7	2.5	3.1	3.2	3.2	1.5
Percentile 25	36	34.9	31	29.9	29	22.6
Percentile 50 (median)	40	39.4	38	37.8	37.5	34
Percentile 75	49.5	48.9	47.9	47.3	47.1	43.6
Interquartile range	13.5	14	16.9	17.5	18.1	21
**Maximum value of mediolateral axis**						
Mean ± standard deviation	53.9 ± 16.6	52.7 ± 12.7	46.9 ± 11.3	48.9 ± 12.4	45.6 ± 12.1	41 ± 10
Kurtosis	2.8	4.6	3.8	3.3	4.5	2
Percentile 25	45.7	41.7	39	38.7	37.3	32.3
Percentile 50 (median)	49.7	48	47.5	47.2	46	43.3
Percentile 75	58.7	55.7	54.8	54.1	53.3	51.3
Interquartile range	13	14	15.8	15.4	16	19
**Mean value of mediolateral axis**						
Mean ± standard deviation	22.8 ± 9	21.9 ± 6.3	20.7 ± 6	21 ± 6.9	19.5 ± 7	15.6 ± 4.2
Kurtosis	2	3.1	4.2	2.4	2.2	2.4
Percentile 25	18.5	17.6	16.6	16.1	14.3	11.2
Percentile 50 (median)	22	21.3	20.8	20.5	19	16.4
Percentile 75	24.6	24.3	23.4	23.4	22.8	21.9
Interquartile range	6.2	6.7	6.9	7.3	8.5	10.7
**Variable**	**G1 (*n* = 187)**	**G2 (*n* = 172)**	**G3 (*n* = 185)**	**G4 (*n* = 192)**	**G5 (*n* = 187)**	**G6 (*n* = 173)**
**Maximum value of anterior-posterior axis**						
Mean ± standard deviation	48.6 ± 14	40.2 ± 10.9	42.8 ± 9.5	39.4 ± 12	40.9 ± 11	33.1 ± 8.6
Kurtosis	2.4	3.4	3.4	2.3	5.4	1.8
Percentile 25	40.3	37.3	33.7	33	28	25.3
Percentile 50 (median)	44	42	40.4	40	36	35.7
Percentile 75	50.3	49.9	48.6	48.1	46.5	44.8
Interquartile range	10	12.6	14.9	15.1	18.5	19.5
**Mean value of anterior-posterior axis**						
Mean ± standard deviation	30.5 ± 9.8	23.5 ± 8	24.8 ± 7.8	22.6 ± 7.2	22.1 ± 6.9	19.3 ± 6
Kurtosis	1.8	3.5	3.5	1.8	1.4	2.3
Percentile 25	22.6	20.5	16.7	16.6	16.1	14.9
Percentile 50 (median)	25.1	23.5	21.6	21.6	21.3	21.2
Percentile 75	28.9	28.1	25.6	25.6	25.5	25.1
Interquartile range	6.3	7.6	9	9	9.4	10.2
**Maximum value of root mean square of accelerations**						
Mean ± standard deviation	85.7 ± 18.2	81.3 ± 19.8	78.9 ± 13.5	72.4 ± 13.5	71.9 ± 9.9	61 ± 13.8
Kurtosis	2.4	3.8	3.4	2.2	2.6	1.4
Percentile 25	72.7	71.7	69	65.8	63.7	47.1
Percentile 50 (median)	77.4	76.9	76.5	73.6	73.2	64.7
Percentile 75	86.9	86.6	84.6	83.6	83.4	73.3
Interquartile range	14.3	15	15.6	17.8	19.6	26.2
**Mean value of root mean square of accelerations**						
Mean ± standard deviation	62.6 ± 15	56.9 ± 14	56.5 ± 11.6	51.5 ± 10.1	50.7 ± 8.8	44.3 ± 11.1
Kurtosis	2.5	3.2	2.6	2.3	2.5	1.4
Percentile 25	54.4	51.5	49.5	46	42.9	34.4
Percentile 50 (median)	58.5	57.4	56.6	56.2	54	49.5
Percentile 75	65.7	64.5	63.1	61.7	59.7	54.5
Interquartile range	11.3	13	13.6	15.7	16.7	20.1

**Table 3 brainsci-10-00832-t003:** Percentiles and descriptive statistics (in seconds) for the duration of the gait test by age group.

Variable	G1 (*n* = 187)	G2 (*n* = 172)	G3 (*n* = 185)	G4 (*n* = 192)	G5 (*n* = 187)	G6 (*n* = 173)
Mean ± standard deviation	15.9 ± 2.3	15.5 ± 2	15.9 ± 1.9	17.1 ± 2.9	16.6 ± 2.1	18.8 ± 3.2
Kurtosis	2	2.4	3.2	4.4	3.5	2.3
Percentile 25	13.7	14.3	14.7	14.7	15.3	16.7
Percentile 50 (median)	14.3	15.2	15.5	16	17	19
Percentile 75	15.7	16.3	17.7	18.3	19.3	21
Interquartile range	2	2	3	3.7	4	4.3

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
