# Peer review of "Percentiles and Reference Values for Accelerometric Gait Assessment in Women Aged 50–80 Years"

_brainsci, 2020, doi:10.3390/brainsci10110832_

Round 1

Reviewer 1 Report

The authors have carried out a very interesting study with relevant results to take into account. However, certain content must be reviewed:  

Abstract:

Line 19: The phrase " To construct the percentile tables and determining the reference values for the accelerometric evaluation of gait in women" needs to express that this is the main objective of the study. For example: This study aimed to construct the percentile...

Line 22: The date of the abstract and of the material and methods is different. In what year was it carried out? 2018 (line 22) or 2019 (line 96)?

Introduction:

Line 70: the word “dynamic” is duplicated.

Line 89: “The reference values thus obtained can then be used to determine whether the gait of a woman is within the normal range. Likewise, the percentiles allow one to make medium- and long-term predictions about functionality and balance, which can be useful for predicting the risk of falling at certain ages.” This information is more appropriate in the discussion or conclusions

Conclusions:  If the authors can further deepen their conclusions and thus emphasize the importance of their observations, this would be a gain for the reader. Specific future applications based on the limitations found or the resulting improvements could then be derived.

Author Response

Dear Editor and Reviewer of Brain Sciences:

Thank you very much for your suggestions and contributions to improve the quality of the manuscript. Following your indications, we respond, point by point, to the reviewers' comments.

In the text, all the modified or added sentences have been written in red to facilitate the correction by the reviewers.

Reviewer 1:

  1. Abstract, Line 19: The phrase "To construct the percentile tables and determining the reference values for the accelerometric evaluation of gait in women" needs to express that this is the main objective of the study. For example: This study aimed to construct the percentile...

The sentence has been rewritten.

  1. Abstract, Line 22: The date of the abstract and of the material and methods is different. In what year was it carried out? 2018 (line 22) or 2019 (line 96)?

The correct date is 2019 and it has been corrected in the text.

  1. Introduction, Line 70: the word “dynamic” is duplicated.

The typo has been corrected.

  1. Introduction, Line 89: “The reference values thus obtained can then be used to determine whether the gait of a woman is within the normal range. Likewise, the percentiles allow one to make medium- and long-term predictions about functionality and balance, which can be useful for predicting the risk of falling at certain ages.” This information is more appropriate in the discussion or conclusions.

The indicated phrases have been removed from the Introduction and are now at the end of the Discussion.

  1. Conclusions:If the authors can further deepen their conclusions and thus emphasize the importance of their observations, this would be a gain for the reader. Specific future applications based on the limitations found or the resulting improvements could then be derived.

The authors have expanded the Discussion and Conclusions emphasizing the importance of the data presented for clinical practice in neurology.

Once again, thank you very much for the time spent and the interest shown in this work; as well as in the positive evaluations you have given of it.

Receive a warm greeting,

The authors.

Reviewer 2 Report

Manuscript ID: brainsci-959011

Manuscript title: Percentiles and reference values for accelerometric gait assessment in women aged 50-80 years

General comments:

This study attempted to create normative values for fall risks in older females through accelerometry during gait. I believe this study has promise and has merit, however, as noted below, there are several concerns I have with the study. I believe if these can be addressed, the study has potential to be publishable. I also highly encourage the authors to edit the short paragraphs and combine them into larger paragraphs. With these, the paper is fairly difficult to read.

Introduction:

In the second paragraph of the introduction, the authors mention that previous fall related studies primarily utilize force platforms and electronic walkways to examine fall risks via center or pressure. However, this is very misleading as CoP is one variable, but it is generally not analyzed with respect to the body. The authors omit a large body of research that further utilizes three-dimensional motion capture systems to understand kinematic characteristics of falls. I highly recommend the authors visit this literature and include it in their introduction as it may assist in framing the introduction better.

The focus of the introduction is primarily on the shortcomings of previous literature and tools used for identifying fall risks. While this is fine, the authors do not provide an overall argument for the purpose of their study. There is a one-sentence paragraph in lines 58-61 supporting their study. I highly suggest the authors revisit the introduction and make a stronger claim for their purpose. I understand the applicability of their study, but general readership may not.

The introduction is very long. I suggest the authors condense it for better readability.

Methods:

The authors describe their recruitment from municipal sports centers. Because of this, I am concerned that this population may not be fully representative of the general population. I suggest that the authors provide rationale for using this population.

I want to applaud the author for recruiting a large sample, very impressive.

The filter cutoff seems arbitrary as there are no citations or analysis on the selection of the cutoff frequency. Please provide justification.

I am confused by the cutoff frequency that was used, was it 30Hz or 20Hz?

The authors state the turns ‘were properly marked’, please provide the methods for identifying these events.

Why are the authors utilizing average acceleration as their primary variable? WOuldn’t instantaneous acceleration provide a greater identification tool? When a fall occurs, it is a discrete timepoint, utilizing average acceleration may not be robust. Furthermore, there are various points in the gait cycle with markedly different acceleration profile, why omit those?

Results:

Beginning in line 251, the authors mention a logistical regression. This is the first time the authors mention this test. If this will be used, it needs to be mentioned in the statistical analyses portion of the methods.

Discussion:

I would caution the authors on stating that these values are representative of the population. The inclusion criteria were fairly stringent, limiting the type of individual who could participate. I believe this statement is premature, which the authors state later in the discussion.

The limitations noted are fairly limited. I suggest the authors take a critical view of their study and include others. I would suggest the restrictive sample as a major limitation since many individuals may not fit into that demographic.

Author Response

Dear Editor and Reviewer of Brain Sciences:

Thank you very much for your suggestions and contributions to improve the quality of the manuscript. Following your indications, we respond, point by point, to the reviewers' comments.

In the text, all the modified or added sentences have been written in red to facilitate the correction by the reviewers.

Reviewer 2:

  1. General comments: This study attempted to create normative values for fall risks in older females through accelerometry during gait. I believe this study has promise and has merit, however, as noted below, there are several concerns I have with the study. I believe if these can be addressed, the study has potential to be publishable. I also highly encourage the authors to edit the short paragraphs and combine them into larger paragraphs. With these, the paper is fairly difficult to read.

The authors appreciate the very positive assessment of our research.

First, to point out that the short paragraphs have been edited and combined throughout the manuscript.

  1. Introduction: In the second paragraph of the introduction, the authors mention that previous fall related studies primarily utilize force platforms and electronic walkways to examine fall risks via center or pressure. However, this is very misleading as CoP is one variable, but it is generally not analyzed with respect to the body. The authors omit a large body of research that further utilizes three-dimensional motion capture systems to understand kinematic characteristics of falls. I highly recommend the authors visit this literature and include it in their introduction as it may assist in framing the introduction better.

The focus of the introduction is primarily on the shortcomings of previous literature and tools used for identifying fall risks. While this is fine, the authors do not provide an overall argument for the purpose of their study. There is a one-sentence paragraph in lines 58-61 supporting their study. I highly suggest the authors revisit the introduction and make a stronger claim for their purpose. I understand the applicability of their study, but general readership may not.

The introduction is very long. I suggest the authors condense it for better readability.

The authors have rewritten the Introduction: we have eliminated and summarized the content previously presented and we have added information about three-dimensional motion capture systems and two sentences that endorse and support the need for this research (final paragraph of the Introduction).

  1. Methods: The authors describe their recruitment from municipal sports centers. Because of this, I am concerned that this population may not be fully representative of the general population. I suggest that the authors provide rationale for using this population. I want to applaud the author for recruiting a large sample, very impressive.

The sample recruitment in municipal sports centers responds to the need to meet the inclusion criteria: (a) engaged in physical activity between 1 and 2 days/week; and (b) walked between 30 and 90 min 4 days a week. At the same time, having recruited participants in municipal sports centers eliminates the socio-economic barrier of having done so in private sports centers.

The authors greatly appreciate the very positive assessment of the work associated with collecting the data from such a large sample.

  1. Methods: The filter cutoff seems arbitrary as there are no citations or analysis on the selection of the cutoff frequency. Please provide justification.

The sentence has been completed with two bibliographic references.

  1. Methods: I am confused by the cutoff frequency that was used, was it 30Hz or 20Hz?

The cutoff frequency used was 30Hz. The error has been corrected.

  1. Methods: The authors state the turns ‘were properly marked’, please provide the methods for identifying these events.

The authors have completed the explanation in the manuscript: “a 50 cm high plastic cone at each end of the run”.

  1. Methods: Why are the authors utilizing average acceleration as their primary variable? Wouldn’t instantaneous acceleration provide a greater identification tool? When a fall occurs, it is a discrete timepoint, utilizing average acceleration may not be robust. Furthermore, there are various points in the gait cycle with markedly different acceleration profile, why omit those?

Although the specific aspects to which you refer are true, it must be taken into account:
- Accelerations during falls were not analyzed in this study. If not, accelerations during NORMAL gait in HEALTHY women.
- Although there are various points in the gait cycle with markedly different acceleration profile, the magnitude of the accelerations at those points depends on the speed of the march. Therefore, averaging the accelerations during three runs of 20 meters in total has been determined to be a test with sufficient validity and reliability:

Leirós-Rodríguez R, Romo-Pérez V, Arce ME, García-Soidán JL. Associations between body composition and movements during gait in women. Rev Int Med Cienc Act Fís Deporte. 2018;18(72):693-707. Doi: 10.15366/rimcafd2018.72.006.

Leirós-Rodríguez R, García-Soidán JL, Romo-Pérez V. Analyzing the use of accelerometers as a method of early diagnosis of alterations in balance in elderly people: A systematic review. Sensors. 2019;19(18):3883-3907. Doi: 10.3390/s19183883.

Leirós-Rodríguez R, Romo-Pérez V, García-Soidán JL, Soto-Rodríguez A. Identification of body balance deterioration of gait in women using accelerometers. Sustainability. 2020;12(3):1222-1231. Doi: 10.3390/su12031222.
- The average of the accelerations in the three axes and their Root Mean Square are the most frequently used variables. Consequently, the reference values ​​provided will be more useful for other specialists already using accelerometry or for comparison with published studies.

  1. Results: Beginning in line 251, the authors mention a logistical regression. This is the first time the authors mention this test. If this will be used, it needs to be mentioned in the statistical analyses portion of the methods.

The Statistical Analysis subsection has been expanded with this aspect.

  1. Discussion: I would caution the authors on stating that these values are representative of the population. The inclusion criteria were fairly stringent, limiting the type of individual who could participate. I believe this statement is premature, which the authors state later in the discussion.

The authors have expanded the Discussion and, immediately after the sentence in which we affirm that “values are representative of the population” we specify the following:

“It should be noted that the reference values obtained are subject to the eligibility conditions of the participants of this work: healthy women, with an active lifestyle and without trauma or orthopaedic conditioners. Set conditions indicate that the results obtained show the expected results in the study of the balance during gait of a healthy adult or older woman.”

  1. Discussion: The limitations noted are fairly limited. I suggest the authors take a critical view of their study and include others. I would suggest the restrictive sample as a major limitation since many individuals may not fit into that demographic.

The authors have expanded the limitations:

“The main limitations of this study are its cross-sectional design, the absence of men from the sample and the lack of data on middle-aged adults. The most important interaction variable is the specific physical activity carried out by each of the participants and the existing variability in their lifestyles, with heterogeneous security as a consequence of the open nature of the inclusion criteria. It should be noted that the strict criteria of inclusion and exclusion selection of the sample employees at the same time which ensure the reliability and validity of the results, limit their generalizability to the total adult population and older women. Finally, the results of this work cannot be applied in assessments which do not follow the measurement protocol used in this research.”

Once again, thank you very much for the time spent and the interest shown in this work; as well as in the positive evaluations you have given of it.

Receive a warm greeting,

The authors.

Round 2

Reviewer 2 Report

I commend the authors for addressing each of my concerns.